# Evaluation of Flathead Grey Mullets (*Mugil cephalus*) Immunization and Long-Term Protection against *Vibrio harveyi* Infection Using Three Different Vaccine Preparations

**DOI:** 10.3390/ijms24098277

**Published:** 2023-05-05

**Authors:** Rosa Strem, Iris Meiri-Ashkenazi, Na’ama Segal, Roberto Ehrlich, Nadav Shashar, Galit Sharon

**Affiliations:** 1Department of Life Sciences, Eilat Campus, Ben Gurion University of the Negev, Eilat 8855630, Israel; rstrem@ocean.org.il (R.S.); nadavsh@bgu.ac.il (N.S.); 2Israel Oceanographic & Limnological Research Ltd.—National Center for Mariculture, Eilat 8811201, Israel; imeiri@ocean.org.il (I.M.-A.); segaln@ocean.org.il (N.S.); roberto.ehrlich@ocean.org.il (R.E.)

**Keywords:** adjuvant, inactivated vaccine, cross-protection, dot-blot assay, multilayer ELISA, fish welfare

## Abstract

In recent years, flathead grey mullets (*Mugil cephalus*) cultured in Eilat (Israel) have been highly affected by *Vibrio harveyi*, showing neurological signs such as uncoordinated circular swimming followed by high mortality rates. Despite the advances in and different approaches to control vibriosis associated with *Vibrio harveyi*, including commercial vaccines, most of them have not succeeded in long-term protection. In this study, we evaluated the effectiveness, long-term protection, and antibody production of three vaccine preparations: heat-killed bacteria (HKB), membrane proteins denaturation (BME PROT), and internal proteins (INT PROT) developed specifically against *Vibrio harveyi* for grey mullets. Our results show that fish immunized with heat-killed bacteria emulsified with adjuvant presented the most effective and long-lasting protection against the bacterium, and a cross-protection against other bacteria from the *harveyi* clade. The effectiveness of each immunization treatment correlated with the levels of specific antibody production against *Vibrio harveyi* in the serum of the immunized fish.

## 1. Introduction

*Vibrio harveyi* is a luminous Gram-negative marine bacterium widely distributed in the environment [1,2]. The organism is a significant pathogen in cultured shrimp [3,4] and bivalves [5], and has been associated with fish diseases [6,7]. Fish infected with *V. harveyi* can have a wide variety of clinical signs, including spiral or erratic movements, darkened melanosis, eye damage or opacity, ulcers, and hemorrhaging around the mouth area and internal organs, among other affections [4].

The virulence of *V. harveyi* depends on the host species [8], doses [9], length of exposure, and age of the host [10]. Several toxins, virulent factors, and virulence regulators are described for this bacterium, including proteases, phospholipases, hemolysins, and bacteriophages, all of which could be important for the pathogenicity [11,12,13,14,15]. However, little is known about the pathogenicity mechanisms of *V. harveyi* in different hosts [16].

The intensification of global aquaculture activities in the last 20 years has led to disease outbreaks, especially due to bacterial infections, resulting in huge economic losses estimated to be more than USD 9 billion per year worldwide [17]. Novriadi [18] reported mass mortalities in different fish and other aquatic organisms, caused by different *Vibrio* species. In countries such as China or Indonesia, vibriosis infections resulted in annual economic losses of around USD 120 million in the aquaculture industry during the 1990s [19].

The flathead grey mullet (*Mugil cephalus*) is a cosmopolitan teleost in tropical, subtropical, and temperate coastal waters around the world´s major oceans [20,21]. The flexibility of this species occupying a varied aquatic environment at different periods of its life cycle, together with its foraging at the base of the food web, enables it to be abundant and reach high biomass in many parts of its distribution range [20]. All those characteristics make the species attractive to fisheries in both freshwater and marine aquaculture [22,23].

Grey mullet from breeding stocks, as well as juveniles and larvae, have been highly affected by *Vibrio harveyi*, showing neurological signs such as uncoordinated circular swimming and oral hemorrhages, causing high mortality rates in cultured populations in the south of Israel [24].

Despite the advances and different approaches to control the spread of the vibriosis associated with *Vibrio harveyi* in different organisms during the last 20 years, not all have been equally successful in preventing losses in the aquaculture industry [25]. Commercial vaccines have been developed against *V. harveyi*, mainly for the shrimp industry and to control fish disease. However, most have not succeeded in long-term protection [16]. The present study evaluated the effectivity, long-term protection, and antibody production of three bacterin preparations developed specifically against *Vibrio harveyi* for grey mullets.

## 2. Results

### 2.1. Fish Mortality Rates Post Challenges with Vibrio harveyi and Bacteriology Analysis

Fish immunized with three different vaccine preparations were experimentally IP infected with live *Vibrio harveyi* or sham-challenged with PBS seven weeks post-initial immunization. Fish from the sham-challenged non-immunized control group exhibited 8% mortality, probably due to the stress of the handling, while fish from the non-immunized control group challenged with live bacteria reached 61% mortality over ten days post-infection (Figure 1A); most mortalities occurred during the first two days. The groups immunized with the three different vaccine preparations did not present any mortality post-challenge with both PBS and live bacteria.

A second challenge with live *Vibrio harveyi* was performed in all fish groups 24 weeks post initial immunization to determine the long-term protection of the fish. All sham-challenged treatment groups (PBS) had a 100% survival rate over the first 10 days post-challenge (results not shown). However, of all the immunized fish groups challenged with live *Vibrio harveyi*, only those immunized with heat-killed bacteria (HKB) presented 100% survival. The control group (non-immunized fish exposed to the live bacteria during the first challenge) showed 11% mortality. The group immunized with the denatured membrane proteins (BME PROT) showed 4% mortality, and the group immunized with the internal proteins (INT PROT) showed 20% mortality over ten days post-infection (Figure 1B).

Bacteriology analysis from the first challenge with live *Vibrio harveyi* was performed on freshly dead fish to confirm etiology. Samples collected from the liver, spleen, kidney, and brain showed negative results for the control group challenged with PBS. In contrast, the bacteriology analysis from the organs collected from the control group challenged with live bacteria was positive for *Vibrio harveyi*. In the case of the second challenge infection, the results from the bacteriology analysis did not recover the same bacterium injected, but two other bacteria belonging to the *harveyi* clade: *Vibrio owensii* and *Vibrio alginolyticus*. These two bacteria commonly occur as co-aggregates in the water during certain seasons [15].

### 2.2. Production and Standardization of Mice Anti-Mullet Antibodies

Mice anti-mullet antibodies were produced since these were not available in the market. The results from the polyacrylamide gel electrophoresis showed that the serum samples preabsorbed onto protein A/G beads included one major protein band of ~75 kDa, which is likely to correspond to the mullet IgM H (heavy chain), and the second, less intense protein band of ~25 kDa, corresponding to the mullet IgM L (light chain; Figure 2A). To evaluate the specificity of the mice anti-mullet Ig, a western blot analysis was developed containing the mullet serum after their absorption to protein A/G beads. The immunoblot revealed a specific response against the protein bands, likely representing the mullet IgM H and IgM L chains, respectively (Figure 2B).

Additionally, all serum dilution titers from the immunized mice (10^4^ to 10^6^) immune reacted, with different intensity levels, with the mullet serum, compared to the same serum dilution titers of the naïve mouse which had no reaction over the mullet serum in a dot–blot assay (Figure 3A). The assay revealed that the 10^5^ dilution of the mice serum provided the best results for further analysis. Different 10-fold fish serum dilutions ranging from 10^3^ to 10^8^ were tested in the ELISA using a 10^5^ dilution of mice anti-mullet antibody serum and naïve mouse serum, as shown in the dot–blot assay. The ELISA OD readings decreased as the dilutions of fish serum increased. The dilution of fish serum selected for future ELISAs was 1:800. The results from the serum of the naïve mouse were consistent with the dot–blot assay, with no reaction over the mullet serum at all dilutions (Figure 3B).

### 2.3. ELISA Analysis of Mullet Anti-Vibrio harveyi Antibodies in Immunized Fish

Serum samples from fish immunized with the different vaccine preparations (HKB, BME PROT, and INT PROT) were assessed for anti-*Vibrio harveyi* antibodies quantification. Serum samples from each of the treatment groups were collected weekly during the experimental trial, and the evaluation of the results was divided into two parts: (1) sampling before the initial immunization injection (day 1) until the sampling before the challenge (week 7), and (2) the evaluation of the production of antibodies from the first challenge infection (week 7) until the last sampling, 18 weeks after the first challenge and before the second challenge (week 24).

The results showed that four weeks after the initial immunization, fish from all groups had a similar increase in specific anti-*Vibrio harveyi* antibody production. The control group, injected with PBS, did not show antibody production at any time point (Figure 4). After the booster injection (week 4), the heat-killed bacteria (HKB) group displayed significantly higher antibody levels compared to the other treatment groups (*p* < 0.05): the internal proteins (INT PROT) and the denatured membrane proteins (BME PROT, Figure 4).

Seven weeks post-immunization, all the treatment groups were either challenged with live *Vibrio harveyi* or sham-challenged with PBS. In the group of fish which were sham challenged, the fish immunized with HKB kept the highest antibody level, followed by the fish from the INT PROT group. On the other hand, the antibody levels of the fish immunized with the BME PROT decreased over time (Figure 5A).

Antibody levels in the group of fish challenged with live *Vibrio harveyi* exhibited a drastic increase in antibody levels after the challenge. In the control group, the antibody levels increased after the challenge with live bacteria but then dropped at 4 weeks post-challenge (week 11; Figure 5B). After 17 more weeks (week 24), the antibodies increased again, probably due to a natural infection, but remained lower than the other treatment groups at all time points. Moreover, the antibody levels from all other treatment groups increased post-challenge and remained high over time at most time points (Figure 5B).

### 2.4. ELISA Analysis to Detect Cross Reactivity against Other Vibrio spp.

After the second challenge with live *Vibrio harveyi*, fish mortalities were associated with two different bacteria from the *harveyi* clade: *V. owensii* and *V. alginolyticus*. Therefore, an ELISA was performed to detect antibody cross-reactivity from the different immunization treatments against these two bacteria. All the fish from the three different treatment groups plus the control did not present high levels of antibodies cross-reacting with the bacterium *Vibrio owensii*, compared to the antibody levels produced against the other two bacteria: *Vibrio harveyi*, and *Vibrio alginolyticus*. However, the fish immunized with the heat-killed bacteria presented high levels of antibodies cross-reacting with the three types of bacteria from the *harveyi* clade (Figure 6). Although the antibody production in the HKB vaccinated group was not the highest against the three bacteria, it was the only group that did not present any mortality during the second challenge infection with *Vibrio harveyi*, resulting in full protection to the fish (Figure 1B).

### 2.5. Evaluation of Heath-Killed Bacteria’s Long-Term Protection, and Antibody Production

Results from the second experimental group of fish immunized with heat-killed bacteria (BAC + PBS) and heat-killed bacteria + adjuvant (BAC + ADJ) showed no mortalities for the group of fish sham-challenged with PBS. On the other hand, mortality rates in the fish challenged with live *Vibrio harveyi* reacted according to the treatments that were exposed to. The control group that was immunized only with PBS resulted in 100% mortality, while the control group that was immunized with PBS + adjuvant (PBS + ADJ) resulted in 89% mortality. In the case of the group of fish immunized with heat-killed bacteria resuspended in PBS (BAC + PBS), the mortality rate reached 40% compared to the group immunized with heat-killed bacteria + adjuvant (BAC + ADJ), which did not present any mortality after the challenge with live *Vibrio harveyi* (Figure 7). Bacteriology analysis from different organs of freshly dead fish from the different treatments had positive results for *Vibrio harveyi*.

In general, results from the multilayer ELISA for the quantification of specific anti-*Vibrio harveyi* antibodies showed a significant difference between the groups immunized with the heat-killed bacteria compared to the control. The group of fish that were immunized with heat-killed bacteria + adjuvant presented significantly higher levels of antibodies throughout the 14 months of the monitoring. During the first eight months from the initial immunization to the booster, the fish were exposed two times to natural infections through the water, resulting in an increase in the antibody levels in all the fish, including the control fish injected with PBS and PBS + ADJ. Those natural infections were observed after running the samples during the ELISA, probably due to a failure in the UV system at the quarantine facilities. No mortalities or clinical signs were observed in the fish during those natural infections. However, after the infection, those fish showed a decrease in their antibody levels (Figure 8). Eight months after the initial immunization, the fish were boosted with their respective vaccine treatments, triggering an increase in the antibody levels, mainly in the group of fish immunized by the heat-killed bacteria + adjuvant, compared to the other groups. One month after the booster injection, as in the months before, the fish were exposed once again to a natural infection, and the antibody levels from all the groups had a drastic increase, yet the antibody levels from the control groups were decreasing over time until the challenge infection with live *Vibrio harveyi* (Figure 8). These results correlate to the mortalities observed after the challenge, where most of the fish from the two control groups died (Figure 7).

## 3. Discussion

Bacterial diseases are considered one of the major limitations to the global aquaculture industry, resulting in high economic losses every year. Nowadays, there is a wide range of available commercial vaccines against bacterial pathogens, mainly targeting specific fish species of economic interest, such as salmon, trout, channel catfish, European sea bass, and sea bream, among others [25]. Moreover, many new vaccines for aquaculture are in development, considering safety, long-term protection, and cost-effectiveness [16].

In the present study, we evaluated the effectivity, long-term protection, and antibody production in juvenile naïve flathead grey mullets immunized by intraperitoneal injections with three *Vibrio harveyi* vaccine preparations: heat-killed bacteria [HKB], denatured membrane proteins [BME PROT], and internal proteins denaturation [INT PROT]. Seven weeks after immunization and booster, fish were challenged with live *Vibrio harveyi* with a dose of 1 × 10^7^ CFU/mL^−1^. The health conditions of the fish were monitored along the experiments. Measurements of weight were taken from five fish per treatment before each sampling; however, there was not a significant difference between the weight of the fish with the different treatments. Our results showed total protection in the immunized fish with the three different vaccine preparations compared to the control group (60% mortality rate). Many studies show that vaccines play a significant role in inducing an immune response and increasing the resistance to diseases in the host´s system [24,25,26,27,28,29,30,31,32], as it was demonstrated in this research.

The ELISA assay results showed a similar correlation between the increase in the anti-*Vibrio harveyi* antibody levels compared with the protection and mortalities observed in each of the immunized groups of grey mullets. After the first challenge, there was a substantial difference in the antibody levels between all the treatments sham challenged with PBS, compared to the group challenged with live *Vibrio harveyi*, as it could be expected. Similar results were obtained in Atlantic salmon (*Salmo salar* L.) vaccinated against *Vibrio salmonicida*, where after challenge infection with live bacteria, the vaccinated groups had no significant difference in the antibody levels and the relative percentage of survival between groups, compared with the unvaccinated fish with lower antibody levels [33].

A second challenge with the LD50 dose of live *Vibrio harveyi* was performed six months after the initial immunization. In terms of mortality rates, the only group that did not present mortalities was the one immunized with the HKB. The results from the quantification of specific anti-*Vibrio harveyi* antibodies are consistent with the cumulative mortalities in each of the groups, showing that fish immunized with the heat-killed bacteria presented not only higher levels of antibodies during all time periods but also exhibited long-lasting protection compared with the other two preparations. These results are consistent with the ones obtained by Crosbie and Nowak [34], where they administered a whole-cell *Vibrio harveyi* vaccine to barramundi (*Lates calcarifer*), creating an immune response to vaccination with demonstrable antibody production. On the other hand, Mohd-Aris et al. [35] developed a live attenuated *Vibrio harveyi* vaccine specific for groupers (*Epinephelus fuscoguttatus*). The vaccinated fish were challenged four weeks post-vaccination with live *Vibrio harveyi* with an LD50 dose of 10^6^ CFU/Fish, resulting only in a 52% relative percentage of survival. Hence, the present results reinforce and justify the increased commercial attraction to inactivated whole-cell vaccines, given their effectiveness in inducing antibody production and long-term protection in fish, as well as their inexpensive costs of production, compared to other kinds of vaccines [16,30].

Bacteriology analyses of freshly dead fish from the second challenge did not recover the original *Vibrio harveyi* injected into the fish but recovered two different bacteria: *Vibrio owensii* and *Vibrio alginolyticus*. Both bacteria belong to the *harveyi* clade and commonly occur as co-aggregation in the water during certain seasons [15]. Co-aggregation describes the formation of bacterial aggregates among different bacterial strains or even species occurring in aquatic bacterial communities or in the intestinal tract of different fish species [36]. These aggregations help the bacteria in the fast adaptation to environmental changes, resulting in an enhanced survival rate, and sometimes help in the increase in the pathogenicity of these bacterial species, evading the host mechanisms of defense [37]. The mortalities observed during our second challenge were probably due to the two bacteria recovered. The results correlate with the increase in antibodies, probably due to a natural infection through the water, before the second challenge with live *Vibrio harveyi*. However, these bacteria are not host-specific, indicating that cross-infections can occur between fish infected with different pathogens and that such diseases are induced by several factors [28].

An ELISA was performed to quantify the antibody levels from fish from the different treatment groups, before the second challenge, against *V. owensii* and *V. alginolyticus*. Fish immunized with the INT PROT and the control group presented the lowest levels of antibodies. These results are consistent with the higher mortalities observed in the fish immunized with the INT PROT bacterial preparation and the control groups (20% and 11% mortality, respectively) following the second challenge. The HKB preparation provided the immunized fish long-term protection against *V. harveyi* and perhaps some degree of cross-protection against the other bacteria from the same clade, as well as the BME PROT (with only 4% mortality) compared to the INT PROT which was less effective. The results could be due to the fact that the HKB and the BME PROT preserved the epitopes binding sites on the surface of the bacterium which are recognized by the antibodies from the host used to neutralize the pathogen [38].

Similar results were obtained by Hettiarachchi et al. [39], in an immersion vaccination trial with two *V. harveyi* vaccines preparation (formalin-killed and heat-killed bacteria) in larvae of cultured shrimp (*Penaeus monodon*) as immunostimulants. β-glucan and peptidoglycan are important immunostimulants increasing the resistance against bacterial infections in different organisms (both components are present in the bacterial cell wall of *V. harveyi*). In our study, this could have acted as well, providing a long-term immunostimulant, resulting in the higher survival of the group immunized by the HKB during the second challenge, and not in the same way in the group vaccinated with the BME PROT, since the dimers from the peptidoglycans are sensitive to the denaturation produced by β-Mercaptoethanol. Furthermore, Mohd-Aris et al. [28] in their review discuss the ability of the outer membrane proteins (OMP) to induce effective cross-species protection against other Vibrio species from the same clade of *harveyi* like *V. alginolyticus* and *V. parahaemolyticus* [40], similar to our results, where the HKB immunized group showed a cross-protection (with no mortalities) against the infections by *Vibrio owensii* and *Vibrio alginolyticus*.

In our research, to evaluate the long-term protection of the heat-killed bacteria emulsified in adjuvant and the effect on antibody production, a second group of juvenile grey mullets was immunized, boosted (8 months after initial immunization), and challenged with live *Vibrio harveyi* (14 months post initial immunization). Results from this challenge showed a 60% survival rate in the group immunized with the heat-killed bacteria, compared to 100% survival in the group vaccinated with the HKB emulsified in adjuvant. These results correlate to the levels of antibody production in each of the vaccinated groups, where the fish immunized with the heat-killed bacteria emulsified in adjuvant presented significantly higher levels of antibodies throughout the 14 months of the experiment, compared to the fish immunized with the heat-killed bacteria + PBS. Many studies have been conducted toward the improvement of vaccine performance such as capturing the antigens in liposomes, or the addition of adjuvants to lead to better immunomodulation and increase protection [16,41,42,43]. Adjuvants are essential elements that increase the efficacy of vaccination practices through many different actions such as stimulators of immune responses [30,44]. Firdaus-Nawi et al. [45] demonstrated the increased effectiveness of killed whole-cell *Streptococcus agalactiae* vaccine added with adjuvant, resulting in 100% survival of red tilapia IP challenged with the live bacteria, compared to 50% survival of fish vaccinated with the killed whole-cell vaccine without adjuvant.

In terms of long-lasting protection, in a literature review of different vaccines applied to different fish species, not many studies report a specific time of protection. There are some studies that report protection of one month such as the case of a red hybrid tilapia immunized with an inactivated *Vibrio harveyi* vaccine [46], while other studies such as the one carried out by Mohamad et al. [47] reported the protection of eight months in seabream and sea bass vaccinated with an inactivated vaccine against different *Vibrio* species. In our research, we managed to show real protection in fish immunized with the heat-killed bacteria vaccine emulsified with adjuvant even 14 months post-initial immunization.

In summary, our results show that the heat-killed bacteria emulsified in adjuvant is a good alternative to prevent vibriosis caused by *Vibrio harveyi* in flathead grey mullets. This vaccine offers an effective low-cost solution to reduce the losses in the aquaculture industry incurred by the disease, in addition to the reduction in antibiotic dependence and the possible residues affecting the environment.

## 4. Materials and Methods

### 4.1. Preparation of Three Different Vaccines

#### 4.1.1. Isolation and Culture of *Vibrio harveyi* Strain RS2016

*Vibrio harveyi* was isolated from the brain of a flathead grey mullet (*Mugil cephalus*) presenting spiral swimming behavior during an outbreak in 2016, in a laminar flow hood, and streaked onto tryptic soy agar (TSA, DIFCO), prepared with 25% sterile seawater (40 ppt), and incubated at 24 ± 1 °C for 48 h. After incubation, the bacterium was inoculated in tryptic soy broth (TSB, DIFCO), prepared with 25% sterile seawater (40 ppt), and incubated for another 24 h at 24 ± 1 °C. Bacterial density was measured at OD600 using a microplate spectrophotometer (PowerWave TMXS, BioTek, Winooski, VT, USA). One part of the bacterial suspension was stored at −80 °C in glycerol (1:1) to keep it in stock for future studies.

#### 4.1.2. Heat-Killed Bacteria (HKB) Preparation

The bacterial suspension in TSB was cultured for 48 h until a density of 1 × 10^9^ CFU/mL^−1^ (Colony-Forming Units), and then heat-killed in a hot water bath (RUBBERMAID 125P, Vi—USA) for 60 min at 60 ± 1 °C. The heat-killed bacteria were harvested by centrifugation (4 °C, 20 min, 3000× *g*; Eppendorf Centrifuge 5417R, Hamburg, Germany) and resuspended in 50 mL of PBS (0.15 M phosphate-buffered saline pH 7.2) up to the initial density [46,47]. This procedure of heat killing was repeated twice to exclude the toxins released during the heating process, and the efficiency of the heat-killed bacterial suspension was verified on TSA plates incubated at 24 ± 1 °C for 48 h. The heat-killed bacterial suspension was subsequently divided into three parts, two of them for the other vaccine preparations, and stored at 4 ± 1 °C.

#### 4.1.3. Denatured Membrane Proteins (BME) Preparation

One part of the heated-killed bacteria was further denatured by the addition of 4% SDS and 10% β mercaptoethanol (BME) at a ratio of 1:1 (*v*/*v*) and dialyzed (MEGA GeBaFlex-tube 20 mL 1 kDa MWCO—Gene Bio-Application Ltd., Yavne, Israel) with repeated changes of PBS at 6 h intervals for 48 h [47].

#### 4.1.4. Internal Proteins Preparation

To obtain the internal proteins from the bacterium, the third part of the heat-killed bacterial suspension was sonicated for 1 hour (20 min cycles with a 10 s pulse on and 20 s pulse off at a temperature of 21 ± 1 °C and Ampl of 30%; Sonics Vibra Cell, Newtown, CT, USA) and then exposed to 20 µL lysozyme (40,000 U/mg, Sigma, St. Louis, MI, USA) + 5 µL DNAse (1 U/µL, ThermoFisher, Waltham, MA, USA) for 20 min at room temperature (RT). Afterward, the suspension with the cell debris was centrifuged (12,000× *g*, 30 min, 4 °C), and the supernatant with the internal proteins was transferred to a new tube, incubated with 5 µL trypsin at 29 ± 1 °C for 4 h, and stored at 4 °C [48,49].

The three bacterial preparations (heat-killed bacteria [HKB], the denatured membrane proteins [BME PROT], and the internal proteins [INT PROT]) were used for fish immunization.

### 4.2. Immunization and Challenge Infections

#### 4.2.1. Fish

A total of 480 clinically healthy juvenile flathead grey mullets (*Mugil cephalus*; ~20 gr) were kept in 100 L aquaria with UV-treated inlet sea water (40 ppt salinity; 100% water exchange, 10 times per day), with aeration at room temperature (RT) during the whole experiment in the quarantine facility at the National Center for Mariculture (NCM), in Eilat, Israel. Water quality parameters (temperature 26 ± 1 °C, pH 7.0–7.5, dissolved oxygen concentrations > 4 ppm, ammonia, and nitrite + nitrate ˂ 0.5 ppm) were monitored weekly and kept at suitable conditions. Fish were fed daily 2% of their body weight of commercial food (BioMar A/S Fish Feed, Brande, Denmark; 2 mm feed with 46/18% protein/fat ratio for marine fish). Fish were left to acclimate for two weeks before starting the experiments and fish health conditions were supervised daily throughout the experiment.

#### 4.2.2. Experimental Design

Fish were randomly divided into four groups in triplicates (40 fish/replicate in 100 L aquarium). Fish from the three treatment groups were immunized intraperitoneally (IP) with 100 µL/fish of the respective *Vibrio harveyi*’ bacterial preparation (Section 4.1) in a concentration of 1 × 10^9^ CFU/mL^−1^, as follows: one group was immunized with HKB, a second group with the BME PROT, and a third group with the INT PROT. A fourth group was injected IP with 100 µL/fish of PBS as a control group (Figure 9).

Three weeks after the initial immunization, fish were boosted with their respective treatment, this time with a bacterial concentration of 1 × 10^7^ CFU/mL^−1^. Six weeks after the initial immunization, the fish from each replicate (twelve aquaria) were divided in two, making the four treatments into eight triplicates (a total of twenty-four aquaria). The first set of treatments was sham challenged (IP) with 100 µL PBS as a control, and the second set was challenged (IP) with 100 µL of live bacteria at the concentration of 1 × 10^7^ CFU/mL^−1^ (LD 50 dose). Six months after the immunization, fish were challenged again: the sham group with 100 µL PBS as a control, and the second group of fish with 100 µL of live bacteria at the concentration of 1 × 10^7^ CFU/mL^−1^ (Figure 9). For both challenge infections with live *Vibrio harveyi*, the bacterium was verified by sequencing of the 16S rRNA, Hemolysin, and Tox R genes. Fish mortalities and clinical signs were monitored and recorded throughout the experiment.

#### 4.2.3. Samples Collection and Analyses

Blood samples from six random fish per treatment (two per replicate) were collected every week for four months. A final sampling was taken six months post-immunization. The timetable for each blood sample collected is as follows (Table 1): T0 (samples collected before first immunization), T1 (one week after immunization), T2 (two weeks after immunization), T0B (four weeks after immunization and before the booster injection), T1B (five weeks after immunization), T2B (six weeks after immunization), T0C (seven weeks after immunization, and before challenge infection), T1C (eight weeks after immunization), T2C (nine weeks after immunization), T3C (ten weeks after immunization), T4C (eleven weeks after immunization), and T5C (twenty-four weeks after immunization). After each of the samplings, blood samples were incubated at 4 °C overnight and then centrifuged at 3500× *g* at 4 °C for 30 min. The serum from each sample was transferred to a new tube and stored at −20 °C for further use in a multilayer ELISA.

Bacteriology analysis was performed on freshly dead fish to confirm the etiology. Samples from liver, spleen, kidney, and brain were collected in a laminar flow hood and inoculated in TSA agar at 24 ± 1 °C for 48 h. Positive samples with bacterial growth were isolated, and DNA samples were extracted from single colonies and were subsequently amplified through PCR analysis of the 16S rRNA gene [50] using the primers 27F (5′-AGAGTTTGATCCTGGCTCAG-3′) and 1492R (5′-TACGGCTACCTTGTTACGACTT-3′), as well as the hemolysin gene (Primers: VH1F 5′-GAG TTC GGT TTC TTT CAA G-3′, VH1R 5′-GTC ACC CAA TGC TAC GAC CT-3′; [51]) and toxR gene (Primers: ToxRF 5′-GAA GCA GCA CTC ACC GAT-3′, ToxRR: 5′-GGT GAA GAC TCA TCA GCA-3′; [52]) specific for *Vibrio harveyi*. PCR products were purified and sequenced at Hy Laboratories Ltd. (Hylabs, Rehovot, Israel).

### 4.3. Evaluation of Long-Term Protection and Antibody Production of the Heat-Killed Bacteria with Adjuvant Preparation

For the long-term protection evaluation, 48 juvenile grey mullets (~40 ± 12 gr) were tagged with a subcutaneous P-tag (ID-100VB, Trovan) with a specific serial number and subsequently divided into four treatment groups (12 fish per treatment). Fish were kept in 100 L aquaria for 14 months at the quarantine facility in the National Center for Mariculture under the same conditions as the fish from the experimental challenges described above. Two groups were vaccinated by IP injection; the first group was immunized with 100 µL of heat-killed bacteria in a concentration of 1 × 10^9^ CFU/mL^−1^ resuspended in PBS. The second group was immunized with 100 µL of heat-killed bacteria (1 × 10^9^ CFU/mL^−1^) emulsified with adjuvant (MONTANIDE^TM^ ISA 761 VG, Seppic S.A., La Garenne-Colombes, France) at a rate of 1:1 (*v*:*v* ratio). For the control groups, one group was IP injected with 100 µL of PBS emulsified with adjuvant at a rate of 1:1 (*v*:*v* ratio), and the fourth group was IP injected with 100 µL of PBS.

Eight months later, fish from the four treatments were boosted with their respective treatments, and 14 months post-initial immunization, two fish from each of the treatments were sham-challenged with 100 µL of PBS, while the rest of the fish from the four treatments were IP-challenged with 100 µL of live *Vibrio harveyi* in a concentration of 1 × 10^7^ CFU/mL^−1^.

Blood samples from all the fish were collected once a month for a 12-month period. A final sampling was taken 14 months post-immunization, before challenging the fish with live bacteria to see the levels of antibodies and protection at the moment of the challenge. After each of the samplings, blood samples were incubated at 4 °C overnight and then centrifuged at 3500× *g* at 4 °C for 30 min. The serum from each sample was transferred to a new tube and stored at −20 °C for further use in a multilayer ELISA. Bacteriology analysis was performed on freshly dead fish to confirm etiology following the methodology described above.

### 4.4. Production of Mice Anti Mullets Antibodies

The production of anti-mullet immunoglobulin in mice was performed according to Sharon et al. [31]. In short, 200 µL blood was withdrawn from the caudal vein of three anesthetized sick mullets (presenting higher levels of antibodies) using 1 mL syringes with a 30× *g* needle. Blood samples were incubated overnight at 4 °C and then centrifuged at 3500× *g* at 4 °C for 30 min; 100 µL of mullet serum was transferred to a new tube and the rest was frozen at −80 °C. The 100 µL of mullet serum was incubated with 40 µL of protein A/G covalently immobilized on Sepharose beads (ABCam, Cambridge, UK) and shaken for 2 h at room temperature (RT). The protein A/G beads were washed three times in PBS and centrifuged at 14,000× *g* for 1 min at RT; the supernatant was discarded, and the pellet with the proteins was resuspended in PBS at a final volume of 200 µL and emulsified (1:1, *v*/*v* ratio) in complete Freund’s adjuvant (CFA, Invivo Gen, Toulouse, France).

#### 4.4.1. Mice Immunization with Mullet Igs

Three 8-week-old BALB/c mice were immunized with 200 µL of mullet Igs preparation (as described above). Blood was collected from a fourth naïve BALB/c mouse as a negative control. A total of three immunizations were applied at 3-week intervals. All mice were initially immunized with mullet Igs from the protein A/G preparation with CFA, while the two booster preparations were with incomplete Freund´s adjuvant (IFA, Invivo Gen, Toulouse, France). Ten days after the third immunization, the mice were anesthetized by a mixture of 2% xylazine (100 mg/kg; Proxylaz^®^) and ketamine HCl (10 mg/kg; USP, Fort Dodge^®^, Iowa), and blood was withdrawn from the facial vein into serum collection tubes (Mini collect^®^ Greiner bio-one Z-serum SepClef Activator). The blood was centrifuged at 3500× *g* at 4 °C for 30 min. The serum was transferred to a new tube and was frozen at −80 °C for posterior use in the multilayer ELISA [31].

#### 4.4.2. Gel Electrophoresis and Western Blot Analysis

Mullet serum (25 µL) was incubated at room temperature for three hours with 10 µL of protein A/G covalently immobilized on Sepharose beads (ABCam, UK). After the incubation period, the protein A/G beads were washed three times in 400 µL of PBS and centrifuged at 5000× *g* for 1 min at RT, and the supernatant was discarded.

For gel electrophoresis, samples were resuspended in PBS and diluted 3:1 in 4x sample buffer (10% SDS, 50% glycerol, 300 mM Tris pH 6.8, 10% β-mercaptoethanol and 0.005% bromophenol blue), boiled for 5 min at 95 °C, and loaded in a precast gradient gel (4–20%, Bio-Rad). The gel was electrophoresed at 150 V, 400 mA for 60 min (PowerPac^TM^ 300, Bio-Rad Laboratories, Inc., Hercules, CA, USA), followed by staining with Coomassie brilliant blue (Roth, Karlsruhe, Germany). Protein bands were quantified by comparison to a standard reference (Precision Plus Protein^TM^ all Blue Standards, Bio-Rad).

Additionally, samples of mullet serum were fractionated by gel electrophoresis and the proteins were transblotted onto a nitrocellulose membrane (0.45 µm, Bio-Rad) by electrophoresis (1 h at 100 V, 250 mA; PowerPacTM300, Bio-Rad). The membrane was stained with Ponceau S (Sigma), scanned and recorded, washed in PBS-T (PBS with 0.1% Tween-20), and blocked in 3% Bovine serum albumin (BSA)/PBS-T + 0.01% sodium azide (Western blot blocking solution) for 1 h at RT. The membrane was incubated on a shaker and washed three times in PBS-T for 5 min each time, followed by incubation with serum (1:2000) from a mouse immunized with mullet Ig from the protein A/G preparation (primary Ab) or with a non-immunized mouse as a control, diluted in blocking solution, for 1 h at RT. Unbound Abs were removed and the membrane was washed as before and incubated with peroxidase-conjugated goat anti-mouse IgG (Sigma) diluted 1:5000 in PBS-T for 1 h at RT. Following the washes, the membrane was incubated with ECL (enhanced chemiluminescence; Thermo Scientific, Rockford, IL) for 1 min, exposed to X-ray film in the dark, and developed by autoradiography.

#### 4.4.3. Dot–Blot Assay

Three different concentrations of the sick mullet serum (5 µL, 10 µL, and 20 µL) were blotted onto a nitrocellulose membrane of 0.25 µm porosity (Bio-Rad Laboratories, Richmond, CA, USA) by suction through a 96-well Bio-Dot microfiltration apparatus (Bio-Rad). PBS was used as a negative control, and the protein A/G binding mullet serum antibodies were used as a positive control. The membranes were blocked in PBST +5% skim milk for one hour at RT, and then three washes were performed in PBST (PBS with 0.05% Tween-20; 5 min per wash). The membranes were exposed for one hour to the primary antibody (anti-mullet mice serum, as well as the non-immunized mouse serum as a control +0.2% Sodium Azide) at the following dilutions: 1 × 10^4^, 1 × 10^5^, and 1 × 10^6^, followed by three washes of PBST (5 min). The membranes were incubated for another hour at RT with a secondary antibody (peroxidase-conjugated goat anti-mouse IgG—HRP, Sigma; 1:5000) in PBST + 1% skim milk. The membranes were washed again and exposed to a chemiluminescent substrate for HRP (ECL SuperSignal^TM^ West Pico PLUS, Thermo Scientific, USA) for five minutes. Images from the blot were captured by GBox luminescent (SYNGENE, USA) for 4 min [31,53,54,55,56,57].

### 4.5. Multilayer ELISA

#### 4.5.1. Standardization of the Mice Anti-Mullet Serum for the ELISA

For the standardization of the enzyme-linked immunosorbent assay (ELISA), a flat-bottom Costar^®^ 96-well assay plate (CORNING, Glendale, AZ, USA) was coated with mullet serum at 10-fold dilutions ranging between 10^3^ to 10^8^. After, the plate was washed three times with PBS-T, and the anti-mullet mice serum and non-immunized mouse serum were added in the concentration of 10^5^ according to the dot–blot assay results.

#### 4.5.2. ELISA Analysis for Anti-Vibrio Harveyi Antibodies in Mullet Serum

A matrix was developed with different concentrations of bacteria and different dilutions of fish serum, which were combined to calibrate the multilayer ELISA.

The multilayer ELISA was carried out based on Sharon et al. [31], with some modifications. Flat-bottom 96-well assay plates were coated with 100 µL per well of heat-killed *Vibrio harveyi* (1 × 10^8^ cells/mL^−1^) and incubated at 4 °C overnight. After three washes of 5 min with PBS-T, 200 µL of blocking solution (1% gelatin in PBS) were added and incubated for 1.5 h at RT. The wells were washed again, as described before, and 100 µL of the fish serum samples (dilution of 1:800 in PBS) were added to the wells and incubated for 3 h at RT. After three washings with PBS-T (5 min each), 100 µL of mouse anti-mullet Igs (diluted 1:10,000 in PBS + 1% Normal Goat Serum) was added to the wells and incubated for 2 h at RT. The wells were washed again, and supplemented with 100 µL of peroxidase-conjugated goat anti-mouse IgG—HRP (Sigma; dilution of 1:1000 in PBS + 1% NGS) for 1.5 h at RT. After three washings, the wells were supplemented with 100 µL of TMB microwell peroxidase substrate (SureBlue^TM^ KPL, Milford, MA, USA) for 20 min at RT in the dark with gentle shaking. Reactions were stopped by the addition of 50 µL of 1 N H_2_SO_4_, and plates were read at 450 nm using a microplate spectrophotometer (PowerWave^TM^XS, BioTek, Winooski, VT, USA). Samples were tested in triplicates and OD values of the negative control (Nonspecific Binding PBS only) were subtracted from all wells.

#### 4.5.3. ELISA Analysis to Detect Cross-Reactivity against Other *Vibrio* spp.

The ELISA was carried out following the methodology described in Section 4.5.2 with some modifications. In this case, the plates were coated with 100 µL per well of heat-killed suspension of three different bacteria: *Vibrio harveyi, Vibrio owensii,* and *Vibrio alginolyticus* (3.5 × 10^8^ cells/mL^−1^), and with 100 µL of fish serum samples taken from the sampling before the second challenge (twenty-four weeks after immunization; diluted 1:200 in PBS).

### 4.6. Statistical Analyses

Statistical analyses were performed with GraphPad Prism software Version 5 (GraphPrism Software, La Jolla, CA, USA). Shapiro–Wilk and Kolmogorov–Smirnov normality tests were conducted to ascertain a normal distribution. Percentage of survival over time was calculated through survival curves using Kaplan–Meier survival analysis and Log-rank (Mantel–Cox) test. For the antibody analyses, the data were evaluated using two-way ANOVA followed by a Bonferroni post-test for multiple comparisons of means among different treatments and periods. The data were presented as mean ± SEM. Statistical significance was determined at *p* < 0.05.

## 5. Conclusions

After challenging twice with live bacterium, the immunized fish from each group showed different levels of protection. The group of fish immunized with the heat-killed bacteria presented the most effective and lasting protection against *Vibrio harveyi*, as well as a cross-protection against other bacteria from the *harveyi* clade. Furthermore, the effectiveness of each of the immunization treatments correlated with the levels of antibody production against *Vibrio harveyi* in the serum of the immunized fish. Additionally, the use of an adjuvant with the heat-killed bacteria vaccine enhances the production of antibodies, giving the fish long-lasting protection from both natural infections and controlled infection, which was performed 14 months post-initial immunization.

The value of this study stands strong considering the relevance of cultured flathead grey mullets for livelihood and human nutrition in many countries around the Mediterranean basin, including Israel. The aquaculture industry is threatened by *Vibrio harveyi*, which has already caused high mortalities and losses in different fish farms within Israel and around the world. The development of a specific vaccine for grey mullets, providing them long-term protection for up to 14 months, against *Vibrio harveyi*, gives fish health professionals insights that could help them maintain and improve the health of the flathead grey mullet aquaculture industry. Furthermore, the benefits of this vaccine in terms of reducing the usage of antibiotics in relation to bacterial diseases result in safer fish products for human consumption and reduce the risk of environmental impact safety issues.

## Figures and Tables

**Figure 1 ijms-24-08277-f001:**
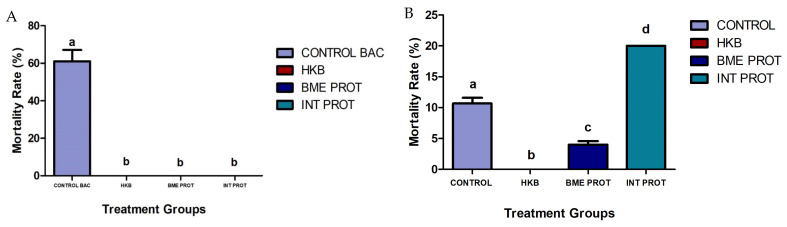
Bar graphs showing cumulative mortality (%) between different treatment groups, over 10 days post challenge infections with live *Vibrio harveyi* (1 × 10^7^ CFU/mL^−1^). The controls from both challenge infections represent the groups challenged with live bacteria. Heat-killed bacteria, denatured membrane proteins with β mercaptoethanol, internal proteins. (**A**) First challenge infection seven weeks post initial immunization: lowercase letters demonstrate significant differences between the control group against the other treatments (*n* = 300. *p* < 0.05). (**B**) Second challenge infection 24 weeks post initial immunization (*n* = 254. *p* < 0.05).

**Figure 2 ijms-24-08277-f002:**
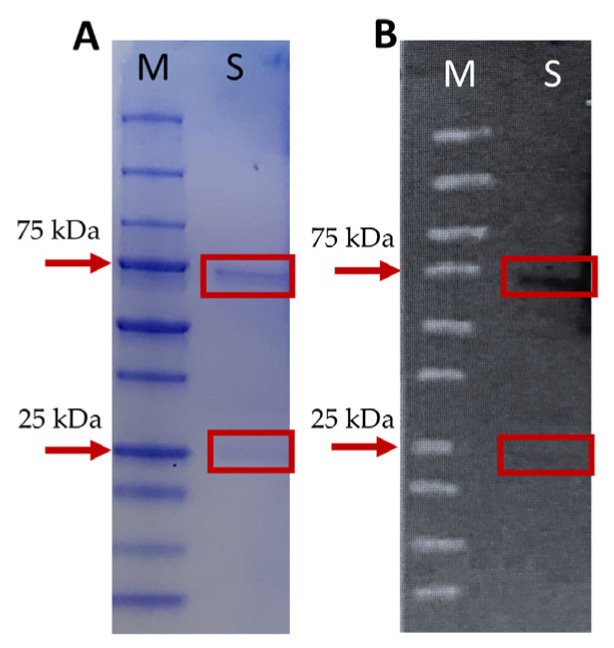
(**A**) Gel electrophoresis stained with Coomassie blue. M: Molecular marker, S: Mullet serum pre-absorbed onto protein A/G beads used for mice vaccination. Arrows show the mass of standard proteins indicated on the left (in kDa). Position of the protein bands representing the IgH and IgL chain (~75 and ~24 kDa, respectively). (**B**) Western blot analysis of mouse anti-mullet sera. Mullet serum pre-absorbed onto protein A/G beads used for mice vaccination was fractionated by SDS-PAGE and proteins were trans-blotted onto nitrocellulose membrane, which was blotted with anti-mullet Ig obtained from an immunized mouse.

**Figure 3 ijms-24-08277-f003:**
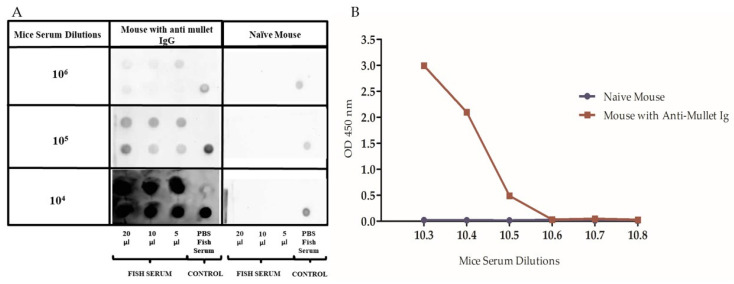
(**A**) Dot–blot analysis. Different volumes of mullet serum (5, 10, and 20 µL) were spotted into different nitrocellulose membranes and treated with three mice anti-mullet antiserum dilution as well as the naïve mouse serum dilution (10^4^, 10^5^, and 10^6^). PBS was used as a negative control, and the protein A/G-binding mullet serum antibodies were used as a positive control. (**B**) ELISA results calibrating different fish serum dilution titers exposed to a 10^5^ dilution from both: a naïve mouse serum and a mouse anti-mullet antiserum.

**Figure 4 ijms-24-08277-f004:**
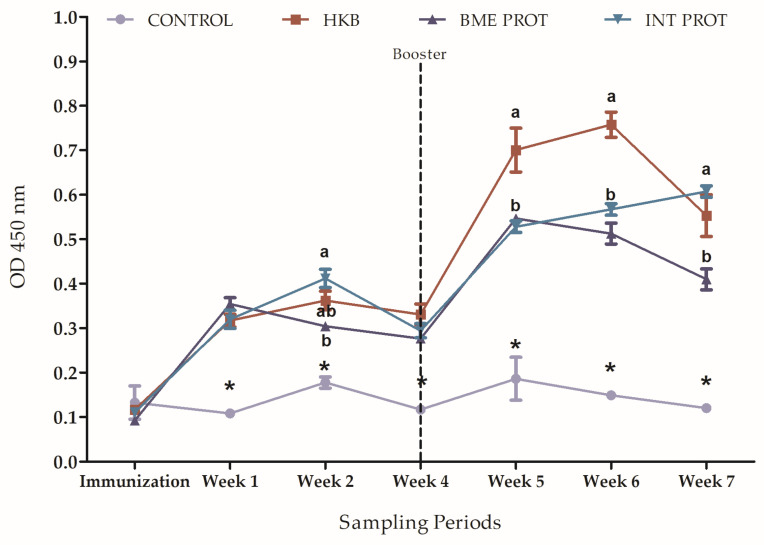
Multilayer ELISA. Quantification of antibody production against *Vibrio harveyi* in each treatment group during the immunization period, from time 0 (Day 1) until the last sampling before the challenge (week 7 post-immunization). Treatments: HKB = Heat-killed bacteria, BME PROT = Denatured membrane proteins with β mercaptoethanol, INT PROT = Internal proteins. Error bars indicate means ± SEM from the three replicates per treatment; sampling times where the error bar is missing means the standard error is smaller than the symbol. Lowercase letters show significant differences between treatment groups and * shows a significant difference between treatment groups and the control (*p* < 0.05).

**Figure 5 ijms-24-08277-f005:**
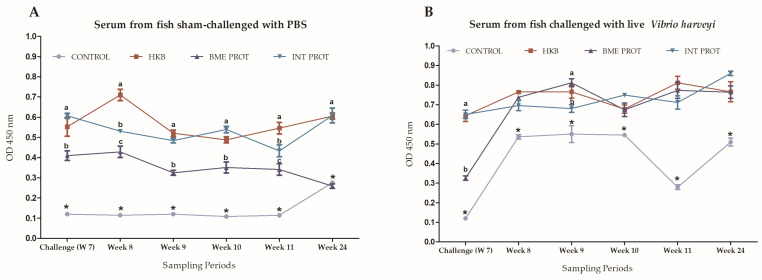
Multilayer ELISA. Quantification of specific anti-*Vibrio harveyi* antibodies from fish immunized with different vaccine preparations (treatment groups). First day of week 7 was the sampling day before the challenge infection, while the sampling on week 24 was the samples taken 17 weeks post-first challenge. Treatments: HKB = Heat-killed bacteria, BME PROT = Denatured membrane proteins with β mercaptoethanol, INT PROT = Internal proteins. Error bars indicate means ± SEM from the three replicates per treatment; sampling times where the error bar is missing means the standard error is smaller than the symbol. Lowercase letters show the significant differences between treatment groups and * shows a significant difference between treatment groups and the control (*n* = 24. *p* < 0.05). (**A**) Antibody levels from the group of fish sham-challenged with PBS. (**B**) Antibody levels from the group of fish challenged with the LD50 dose (1 × 10^7^ CFU/mL^−1^) of live *Vibrio harveyi*.

**Figure 6 ijms-24-08277-f006:**
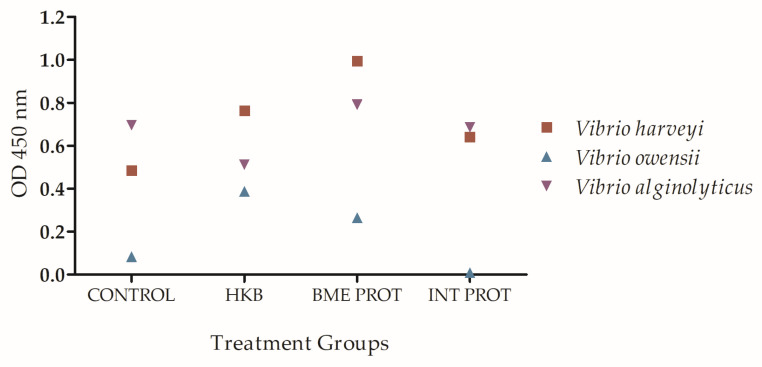
Multilayer ELISA. Quantification of specific antibody levels in fish serum twenty-four weeks post immunization with the three vaccine preparations against three different bacteria: *Vibrio harveyi*, *Vibrio owensii*, and *Vibrio alginolyticus*. Treatments: HKB = Heat-killed bacteria, BME PROT = Denatured membrane proteins with β mercaptoethanol, INT PROT = Internal proteins. Error bars indicate means ± SEM; however, standard error was smaller than the symbol; therefore, error bars are not shown.

**Figure 7 ijms-24-08277-f007:**
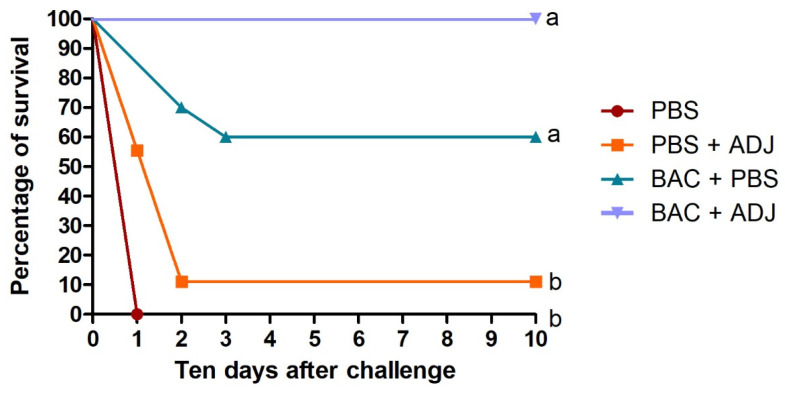
Graph showing fish survival rate from different treatment groups over a 10-day period (%), post-challenge with live *Vibrio harveyi* (1 × 10^7^ CFU/mL^−1^). The control groups were injected with PBS and PBS + adjuvant (PBS + ADJ), and the immunized groups were injected with the heat-killed bacteria (BAC + PBS) and the heat-killed bacteria + adjuvant (BAC + ADJ). Lowercase letters represent a significant difference between the groups that were immunized with the heat-killed bacteria, compared to the control groups injected with PBS (*n* = 48. *p* < 0.05).

**Figure 8 ijms-24-08277-f008:**
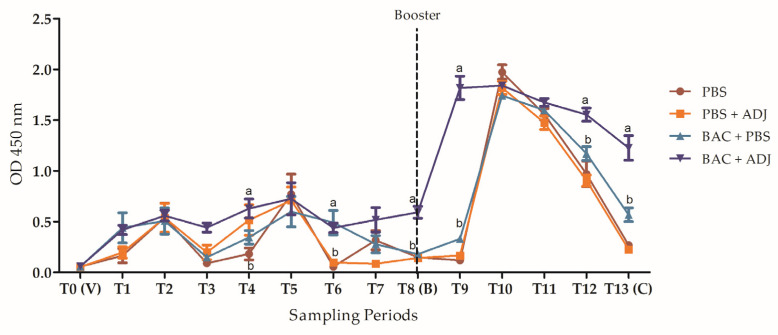
ELISA analysis, quantification of antibody production against *Vibrio harveyi* in each of the treatment groups from the immunization (T0) until the last sampling before the challenge (T13–14 months post-immunization). Serum samples were taken monthly. Treatments: PBS, PBS + adjuvant (PBS + ADJ), heat-killed bacteria resuspended in PBS (BAC + PBS), and heat-killed bacteria + adjuvant (BAC + ADJ). Abbreviations: V—Vaccination, B—Booster or second vaccination, and C—Challenge infection. Sampling times where the error bar is missing mean the standard error is smaller than the symbol. Lowercase letters show significant differences between treatment groups (*n* = 48. *p* < 0.05).

**Figure 9 ijms-24-08277-f009:**
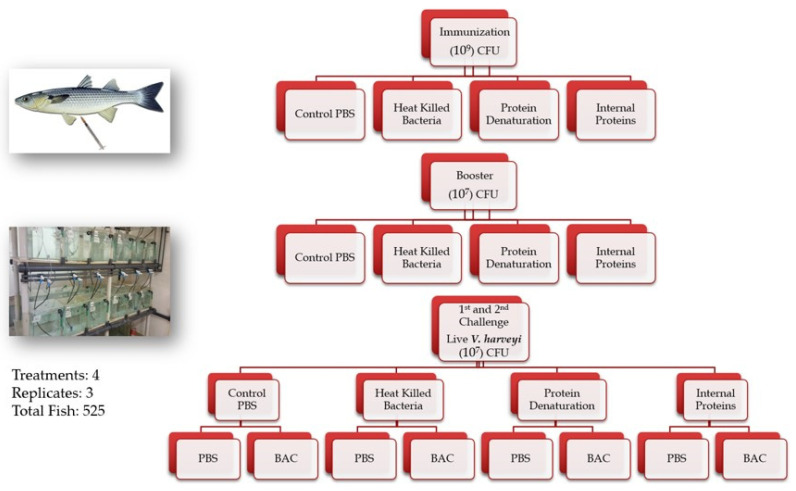
Diagram showing the experimental setup of the four treatment groups including vaccination, booster, and challenge to the fish by IP injection.

**Table 1 ijms-24-08277-t001:** Timetable of the immunization, booster, challenges, and blood sampling.

Chronology	Number of Bleedings	Time (Weeks)
Immunization	Bleeding Time 0 (T0)	0
	Bleeding Time 1 (T1)	1
	Bleeding Time 2 (T2)	2
Booster	Bleeding Time 0 (T0B)	4
	Bleeding Time 1 (T1B)	5
	Bleeding Time 2 (T2B)	6
First Challenge	Bleeding time 0 (T0C)	7
	Bleeding time 1 (T1C)	8
	Bleeding time 2 (T2C)	9
	Bleeding time 3 (T3C)	10
	Bleeding time 4 (T4C)	11
Second Challenge	Bleeding time 5 (T5C)	24

## Data Availability

Not applicable.

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
