# Peer review of "Evaluation of Flathead Grey Mullets (Mugil cephalus) Immunization and Long-Term Protection against Vibrio harveyi Infection Using Three Different Vaccine Preparations"

_ijms, 2023, doi:10.3390/ijms24098277_

Round 1

Reviewer 1 Report

Comments to AUTHOR:

Your articles provide relevant information about the Long-Term Protection Against Vibrio harveyi Infection Using Three Different Vaccine Preparations. However, the poor drafting of texts is one of the major causes of the decline in the value of the information presented. So, I think this article should be revised by following suggestions.

Specific recommendations

Major questions:

1.      Line 65. The sentence “reached 61% mortality rate” is not match Figure 1A, please check the raw data.

2.     Line 68: The sentence “…with both: PBS and live bacteria.”. But in the above sentence, you said “… (injected with PBS)exhibited 8% mortality”, what do you want to express about whether PBS has an influence on survival rate?

3.     Lines 69- 70: Seeming not to have correlation data in this section.

4.     Your experiment has lasted long weeks, why do you just exhibit 10 days of data?

5.     Line 105: Why just have two bands in Figure 2A from the serum sample? And then Figure 2A seeming not in the gel, rather than on the PVDF membrane.

Lines 144-148: The sentences were hard to understand.

Minor questions:

1.      Line 62: Species name should be in italics. please check the species name in this manuscript.

2.      Line 109: antisera or antimullet?

3.      Line 153: Marked significant change P value.

4.      Line 272: bacterial preparations or vaccine preparations?

5.      Line 434 and 446: the unit cell/ml-1 and CFU/ml-1 are not correct, please revise.

Moderate editing of English language

Author Response

Reviewer 1

The answer for each of the comments and suggestions is written in red color after each of the comments.

Reviewer 2 Report

The manuscript entitled “Evaluation of Flathead Gray Mullets (Mugil cephalus) Immunization and Long-Term Protection Against Vibrio harveyi Infection Using Three Different Vaccine Preparations” addresses a topic of great importance for the development of aquaculture of the species M. cephalus. The authors present a clear and objective text. A detailed and reproducible methodology. Robust and easy-to-understand results that will strengthen research aimed at solving the problem caused by pathogenic bacteria in the evaluated species.

General observations:

The authors present a clear, objective and contextualized introduction about the need to develop a vaccine against the bacteria Vibrio harveyi. A suggestion to strengthen this topic is the inclusion of data on loss in aquaculture caused by this bacterium (if any); The authors mention that there is a need to produce long-term protection, it would be interesting to insert information about the protection time already registered for vaccines produced for fish or for the evaluated species.

The results are very detailed about the long-term effect of the vaccine for the animals. To strengthen these results (if possible), I suggest that the authors include information related to the zootechnical performance of the animals throughout the experiment. I believe that in addition to safety against bacteria and non-mortality, it is important to know the effects on animal growth. Are there any positive or negative relationships? It's worth the discussion.

Specific suggestions:

Line 26: Avoid keywords that are already present in the title of the manuscript;

Line 33: Check the term “variety of symptoms”, fish does not show symptoms, I suggest using the term “clinical sign”.

Line 35: Avoid using "etc" in scientific texts. Be specific.

Line 72: For results not shown, I suggest they be included as supplemental material.

Congratulations to the authors for the excellent work.

Author Response

Reviewer 2

The answer for each of the comments and suggestions is written in red color

Reviewer 3 Report

The ms “Evaluation of Flathead Grey Mullets (Mugil cephalus) Immunization and Long-Term Protection Against Vibrio harveyi Infection Using Three Different Vaccine Preparations” by Rosa Strem et al is a nice and interesting contribution to the special issue on fish immunology: the authors display a wide array of immunological techniques to demonstrate that different preparations of Vibrio harveyi elicit immune responses, and that this method has significant potential to be applies practically to the aquaculture of flathead grey mullet. This makes this a fine contribution to the journal.

The paper is generally well written, but I would recommend that the authors carefully revise the text to make it easier to follow without having to read the Materials and Methods section – something which should be possible if the journal allows the results to immediately follow the introduction. For instance, it would be very helpful to state right from the start that experimental infection is via intraperitoneal injection; that sham-infection with PBS was applied to ALL experimental groups, and is not a further experimental group by itself; etc. I enclose an annotated version of the text, which I hope helps the authors during revision.

A couple of strange/odd things could be addressed in the discussion, as well. For instance, the fact that in the second challenge no significant differences in mortality were detected between control and immunized groups; the fact that controls had very high titres of anti-V. harveyi antibodies without having been exposed to the bacteria; or the fact that fish became “naturally” infected in a quarantine installation, which one would assume to be rather “aseptic”. None of these observations demerit the value of the study, but all of them should be discussed.

Author Response

Reviewer 3

The answer for each of the comments and suggestions is written in red color

Reviewer 4 Report

General viewpoint:

The main purpose of this experiment was the evaluation of the effectiveness, long-term protection, and antibody production of three vaccine preparations. The experimental design was comprehensive and the diversity of control groups is properly included in different trials. However, I could not find a description of the objectives of each control group in the Materials and Methods section. Therefore, I suggest you provide an explanation of why each control is included. Also, I think the way used to explain the content, especially in the results section, is very confusing and vague. The results should be written in such a way that the reader can easily understand them. In this case, the authors should simplify and summarize the results and use more appropriate diagrams/ graphs.

Major comments:

From my point of view, the authors have to to take some tissue samples to analyse the expression of immune genes and cell markers such as IgT, IgM, CD4, CD8α, IL-1β to validate their findings regarding the efficacy of vaccines and antibody production. Therefore, I would like the authors to explain whether there is a specific reason for the lack of this information.

Minor comments:

1-     1- In the first part of the abstract (lines 11-18) there is additional information that should be included in the introduction. Delete some lines to keep your abstract shorter.

2-     Line 61 : add ‘Bacteriology analysis‘ to the subheading (Fish Mortality Rates Post Challenges with Vibrio harveyi and bacteriology analysis).

3-     In figures 1 and 2 the data is not clearly presented. Use another chart type, such as a column chart.

4-     In figure 1A: delete BME PROT and INT PROT from the legend.

5-     Figure 1B: which control is that? PBS or BAC.

6-     Where are the ~25 kDa protein bands representing the mullet IgM L chain in Figure 2? I can not see any specific bands there.

7-     In the results section, you have to write only the results. The description of the methods used for each trial have to be transferred to material & method section. For instance, delete the sentences between lines 98-102 as well as lines 211-218.

8-     In figure 4.  significant differences between treatment groups are shown by lowercase letters. Have you checked to see if there is a significant difference in antibody production between the first and second doses of any vaccine? Do you believe a second dose particularly for HKB vaccine is essential to provide maximum immune protection in this species?

9-     Line 162: Which statistical analysis was performed to show a significant difference between treatment groups and the control?

10-  Line 272: Be consistent in writing. Do not repeat the full name after giving the abbreviation of the name of the vaccines. To avoid that, check the entire version. For example, in lines 324, 326, write the abbreviation of internal proteins vaccine.

11-  Line 296. If you have performed long-lasting protection trial only for the fish immunized with the  heat-killed bacteria (as you described in section 4.3) how would you compare it to the other two preparations and conclude that this group showed long-term protection?

12-  What does it mean ‘Production of Three Different Vaccine Preparations’. Write : Preparations of Three Different Vaccines.

13-  Line 384: What do you mean by posterior use? Do you mean future studies?

14-  Line 395: instead of ‘further use‘ write exactly where you are going to use the heat-killed bacterial suspension.

15-  Line 499: Explain why the final sampling was done before the fish were challenged with live bacteria and not compared to the level of antibody production after the fish were challenged with live bacteria.

16-  Line 530: Instead of writing ‘further use’ name the the analyzes in which you used the serum.

17-  Line 605:  Instead of the method described above, write the section number (4.5.2).

Author Response

Reviewer 4

The answer for each of the comments and suggestions is written in red color

Round 2

Reviewer 4 Report

All comments are included in the manuscript. For the 11th comment, I would like to emphasize, since you have performed the long-lasting protection trial for the fish vaccinated with the heat-killed bacteria not the other vaccines, you should remove ‘compared with the other two preparations’ in terms of long-lasting protection.